Using search trends to analyze web-based users’ behavior profiles connected with COVID-19 in mainland China: infodemiology study based on hot words and Baidu Index

Jiang Shuai 1
You Changqiao 2
Zhang Sheng 1
Chen Fenglin 1
Peng Guo 1
Liu Jiajie 1
Xie Daolong 1
Li Yongliang 1608956268@qq.com 1
Guo Xinhong gxh@hnu.edu.cn 1
College of Biology, Hunan University , Changsha , Hunan Province , China
NanHua Bio-medicine Co.,Ltd. , Changsha , Hunan , China
Teh Cindy Shuan Ju
Electronic publication date: 2022 Nov 9
Publication date: 2022
Volume: 10
Electronic Location ID: e14343
Received 2022 Aug 17; Accepted 2022 Oct 14
Copyright: ©2022 Jiang et al.
Copyright year: 2022
Copyright holder: Jiang et al.
License: This is an open access article distributed under the terms of the Creative Commons Attribution License, which permits unrestricted use, distribution, reproduction and adaptation in any medium and for any purpose provided that it is properly attributed. For attribution, the original author(s), title, publication source (PeerJ) and either DOI or URL of the article must be cited.
License URL: https://creativecommons.org/licenses/by/4.0/

Keywords: Behavior profiles, COVID-19, Mainland China, Hot words, Baidu index

Funding: Key Research & Development Project of Nanhua Biomedical Co., Ltd H202191490139 National Natural Science Foundation of China 31872866 China Postdoctoral Science Foundation 2021M701160 This research was supported by grants from the Key Research & Development Project of Nanhua Biomedical Co., Ltd (No H202191490139), the National Natural Science Foundation of China (No 31872866), and the China Postdoctoral Science Foundation (No 2021M701160). The funders had no role in study design, data collection and analysis, decision to publish, or preparation of the manuscript.

==============================
Background

Mainland China, the world’s most populous region, experienced a large-scale coronavirus disease 2019 (COVID-19) outbreak in 2020 and 2021, respectively. Existing infodemiology studies have primarily concentrated on the prospective surveillance of confirmed cases or symptoms which met the criterion for investigators; nevertheless, the actual impact regarding COVID-19 on the public and subsequent attitudes of different groups towards the COVID-19 epidemic were neglected.

Methods

This study aimed to examine the public web-based search trends and behavior patterns related to COVID-19 outbreaks in mainland China by using hot words and Baidu Index (BI). The initial hot words (the high-frequency words on the Internet) and the epidemic data (2019/12/01–2021/11/30) were mined from infodemiology platforms. The final hot words table was established by two-rounds of hot words screening and double-level hot words classification. Temporal distribution and demographic portraits of COVID-19 were queried by search trends service supplied from BI to perform the correlation analysis. Further, we used the parameter estimation to quantitatively forecast the geographical distribution of COVID-19 in the future.

Results

The final English-Chinese bilingual table was established including six domains and 32 subordinate hot words. According to the temporal distribution of domains and subordinate hot words in 2020 and 2021, the peaks of searching subordinate hot words and COVID-19 outbreak periods had significant temporal correlation and the subordinate hot words in COVID-19 Related and Territory domains were reliable for COVID-19 surveillance. Gender distribution results showed that Territory domain (the male proportion: 67.69%; standard deviation (SD): 5.88%) and Symptoms/Symptom and Public Health (the female proportion: 57.95%, 56.61%; SD: 0, 9.06%) domains were searched more by male and female groups respectively. The results of age distribution of hot words showed that people aged 20–50 (middle-aged people) had a higher online search intensity, and the group of 20–29, 30–39 years old focused more on Media and Symptoms/Symptom (proportion: 45.43%, 51.66%; SD: 15.37%, 16.59%) domains respectively. Finally, based on frequency rankings of searching hot words and confirmed cases in Mainland China, the epidemic situation of provinces and Chinese administrative divisions were divided into 5 levels of early-warning regions. Central, East and South China regions would be impacted again by the COVID-19 in the future.

Introduction

As the most prevalent widely infectious disease in the 21st century, coronavirus disease 2019 (COVID-19) seriously threatened the safety of human lives and properties (Lau et al., 2020). In mainland China, the population has stood at nearly 1.4 billion and the average annual passenger volume has reached 600 million in 2021 (Lau et al., 2020; Zhao et al., 2021). Since the first case of COVID-19 was confirmed on December 2019 in Wuhan, China (Lai et al., 2020), researchers have released the whole genomes, protein structure (Akhand et al., 2020) and genetic lineage information (Potdar et al., 2021) about the causative pathogen (Severe acute respiratory syndrome coronavirus 2, SARS-CoV-2) for the COVID-19 (Lau et al., 2020). Mechanisms of SARS-CoV-2 transmission and pathogenesis have also been summarized with clinical research studies for improving public protection awareness (Harrison, Lin & Wang, 2020; Rader et al., 2021; Yasin, Grivna & Abu-Zidan, 2021). In underdeveloped areas with alternate health beliefs and access to medical care, patients were less inclined to seek medical help from Chinese government agencies (Cai et al., 2021), thus causing misleading statistics. Hence, the real-time attitudes and related problems of different groups of patients were easily ignored owing to the lack of national surveillance and corresponding epidemiological reporting systems.

Convenient and real-time services from online big data infodemiology platforms have become a better choice for the public to query public health issues (Eysenbach, 2009). Users had access to these platforms (e.g., GISAID, NCBI, WHO (Zhou et al., 2020), Centers for Disease Control and Prevention (CDC), European Centre for Disease Prevention and Control (ECDC) and Chinese Centers for Disease Control and Prevention (CCDC), etc.) for reading the news and blogs regarding COVID-19 (Zhou et al., 2020). In terms of statistics from the China Internet Network Information Center, the Internet penetration rate and the scale of netizens in mainland China have reached 74.4% and 1,051 million people respectively by June 2022 (CNNIC, 2022). Furthermore, based on the data from China Statistical Yearbook compiled by National Bureau of Statistics of China, the number of mobile internet users have been up to nearly 13.5 million people by the end of 2020 (STATS, 2021). As the most widely used search platform in mainland China, Baidu accounted for 93% of service usage and 92% of search volume data on the whole network (Wei et al., 2021). The Baidu Index (BI) based on the Baidu search platform integrated the temporal, age, gender and geographical information of Baidu’s visitors, thus researchers got the characteristic information of users when inputting single or combined entries on BI (Fang et al., 2021). Additionally, researches conducted by Wang et al. (2020a), Wang et al. (2020b) and Huang et al. (2020) have shown that BI was suitable to construct the behavior portraits of users between different groups, contributing to propounding the insights into the health care concerns and guiding the tracking of common interests of the public. In the future, more people will consider the big data platforms as the optimal channels for inquiring epidemic situation and confirming symptoms (Dreher et al., 2018).

Hot words were known as buzzwords or popular words, which arose in network terms and relied on search engines and platforms as carriers (Xu et al., 2017). Hot words also illustrated the current social phenomena and the popular information concerned by the public in one period of time (Xu et al., 2017), so they spread faster and wider on Internet compared with official statistics (Fang et al., 2021; Xu et al., 2019). The rapid development of big data platforms was vital to analyze users’ portraits by hot words (Xiang et al., 2020). Google Trends has been successfully practiced in analyzing Malaria (Soko, Chimbari & Mukaratirwa, 2015), flu (Kandula & Shaman, 2019) and COVID-19 (Shen et al., 2020) and surveilling routes of transmission. In mainland China, more than 770 million users have actively used the search services provided by Baidu, and 63.16% of the service content involved health and symptom queries by 2021 (Wei et al., 2021). Therefore, it was more feasible to use BI to collect hot words and analyze characteristic behavior profiles of the public.

Numerous mathematical models have been set up to analyze individual or the public behavior patterns during the COVID-19 pandemic (Jewell, Lewnard & Jewell, 2020). These models have focused predominantly on the correlation between clinical data of COVID-19 and a certain factor such as mental health problems in cases (Hossain et al., 2020), traffic volume (Yasin, Grivna & Abu-Zidan, 2021) or air traffic (Lau et al., 2020). For some multi-factor models, their flexibility and accuracy would be greatly reduced due to the repeated testing of the dataset and lack of ample data from platforms (Rader et al., 2021; Fang et al., 2021). Besides, a few models depended on sophisticated algorithms such as Euler equation (Foy et al., 2021) and spatial matrix (Hossain et al., 2020) to enhance the precision. Therefore, the problems of monotonous experimental factors, complex algorithms, incomplete data and poor compatibility could result in unstable, unacceptable and unrepeatable results (Jewell, Lewnard & Jewell, 2020; Hossain et al., 2020).

The chief objective of this study was to assess the validity of reflecting and monitoring the impact of COVID-19 on the people in different age groups, gender groups and regions and their behavior patterns with hot words and BI. As the first research based on the big-data platforms to extract hot words for building a monitoring COVID-19 model, we collected abundant hot words reflecting different public social interests, removed ambiguous data and reasonably enlarged estimation range. Some uncomplicated and precise algorithms such as Baidu Index daily average (BIDA) (Yang et al., 2017) and year-on-year growth rate (YoY +%) (Pei, De Vries & Zhang, 2021) were used to screen hot words and create a Chinese-English bilingual table (Wei et al., 2021). Users’ portraits described by BI finally verified the effectiveness of hot words through the correlation analysis. When proposing models relevant to decision-making in epidemic prevention and control, providing convincing and intelligible evidence for health management departments and medical practitioners promoted the formulation of policies and the implementation of measures (Hossain et al., 2020).

This article was divided into three parts. The first part dealt with the screening of hot words which were finally summarized in the bilingual table. The second part analyzed the correlation of temporal distribution between hot words and epidemic severity. The third part analyzed distribution characteristics (gender, age and geographical) of various groups according to hot words and assessed the effect of monitoring COVID-19 based on above results.

Materials & Methods

The selection and double-level infodemiology classification of hot words

In this study, we extracted the words appearing on the web pages (Table S1) of GISAID, WHO and NCBI. Considering linguistic habits, some common proper nouns and semantic words should be eliminated by the rules in Table 1. Then, we sorted them by word frequency (Table S2) with Python codes (code S1 in Supplemental Files) and screened out top 5% of total words as the initial hot words.

Table 1 The rules for eliminating hot words and corresponding examples.

Eliminating types of hot words	Examples	
Complex abbreviations of organizations and drugs	ACT (Access to COVID-19 Tools), PHEIC (public health emergency of international concern)	
Modal and auxiliary verbs	Can, Do	
Names	Peter, David	
Some special invalid words	Republic, Peoples	
Adjectives and adverbs are adjusted to nouns with similar meanings	Public, Global	
Figures	Years from 2000 to 2022	

Given word classification strategies applied to the WHO’s timeline of events (WHO, 2021), all initial hot words were defined as “subordinate hot words” (the secondary taxon). The typical infodemiology features (Wei et al., 2021) containing in above subordinate hot words were summarized and defined as “domains” (the primary taxon). Subordinate hot words and domains were interpreted into corresponding Chinese entries by referring to translation and interpretation recommendations from CDC and CCDC pages (Table S1) for retrieving on BI.

Because COVID-19-unrelated hot words disturbed the timeliness and sensitivity of epidemic surveillance, we calculated the BIDA of all subordinate hot words in different periods (2019/12/01–2021/11/30 and 2017/12/01–2019/11/30) and their corresponding YoY +% values. Taking 5% as the benchmark, one hot word or domain was related to COVID-19 when its YoY +% exceeded the benchmark (the BIDA of one domain was summed by that of related subordinate hot words). Meanwhile, Chinese entries without search results on BI were also excluded.

Temporal correlation analysis

The number of COVID-19 cases and deaths directly reflected epidemic severity. BI values were calculated by the filtering and term weight adjustment algorithms instead of simply accumulating search volume (Guo, Zhang & Wu, 2021), which resulted in a weak temporal correlation between BI values of subordinate hot words (domains) and epidemic severity. To solve the above problem, this study replaced BI values with the occurrence frequency of subordinate hot words to explore the temporal relationship (Wang et al., 2021). The occurrences of subordinate hot words (domains) was defined as their occurrence frequencies of peak BI values (kurtosis >3, critical value of leptokurtic and fat-tailed distribution (Lenart, Pajor & Kwiatkowski, 2021) to compute their sum of occurrence frequencies per week. Additionally, our study took “Month” as the minimum time unit of temporal relationship analysis instead of “Day” for avoiding possible data redundancy problems (Garland, James & Bradley, 2014) and exploring the secular trends of domains.

Characteristic profiles analysis

We collected statistics about the proportions of gender and age distribution and the weighted rankings (results only showed the top ten provinces) of geographical distribution through retrieving subordinate hot words and domains on BI. Moreover, we collected the cumulative cases (2019/12/01–2021/11/30) of all provinces from National Health Commission of the People’s Republic of China (NHCPRC) (NHC, 2021) for constructing geographical distribution characteristic values (GDCV) of provinces. Given extreme values existing in cumulative cases of a few provinces (Hubei province: 68311; Tibet province: 1), case numbers should be normalized via exponential normalization (nonlinear curve fitting model, P value <0.05) (Barakat, Khaled & Rakha, 2020; Wei & Hutson, 2013) with R language for reducing the impact of extreme values on confidence interval (Krebs et al., 2018). After acquiring high-quality curve fitting (coefficient of determination, R2 ≈1), case numbers in provinces were regarded as the independent variable and substituted into the model to obtain the normalized GDCV of provinces and subordinate hot words. Then, we estimated the confidence interval of GDCV of the top ten provinces where the outbreaks may lead to an explosion in cases (P value <0.05). Based on width of the confidence interval (Wei & Hutson, 2013), all possible occurrence frequencies of COVID-19 outbreaks in provinces were obtained with Python codes (code S2 in Supplemental Files). We evaluated the early-warning levels for the provinces and Chinese administrative divisions (Northeast, North, East, Central, South, Northwest and Southwest China) (Wei et al., 2021) based on the rankings of occurrence frequencies of all provinces. The detailed flowchart of forecasting geographical distribution was shown in Fig. 1.

Figure 1 The flowchart of forecasting geographical distribution based on parameter estimation.

Results

Establishment of hot words table

After two-rounds of hot words screening (linguistic habits and YoY +%) and double-level hot words classification (subordinate hot words and domains) (Table S3), the bilingual table including 6 domains and 32 subordinate hot words was established (Table 2). Only 6 subordinate hot words’ BIDA (COVID-19, SARS-CoV-2+, Coronavirus, USA, China, Wuhan) in two domains (CR and T) exceeded 10,000 (Fig. 2), and YoY +% of “COVID-19”, “SARS-CoV-2+” and “Coronavirus” even exceeded 4500%. Compared with some subordinate hot words with low YoY +% (Fatigue, Clinical, Government+, Organization+, Commission, Report, 9.59%, 5.57%, 5.68%, 5.65%, 8.97%, 9.77%), above 6 subordinate hot words with high YoY +% could better reflect the tremendous influence of COVID-19 on the public.

Table 2 The list of domains and subordinate hot words.

Domains and abbreviations	Subordinate hot words and abbreviations	
COVID-19 Related (CR)	Coronavirus, COVID-19, Infection/Transmission (Infection +), SARS-CoV-2/Novel coronavirus (SARS-CoV-2 +), Variant	
Government (G)	Assembly, Commission, Expert, Foundation, Government/Governments (Government +), Measure/Initiative (Measure +), Response, Studies/Research (Studies +), Support, Organization/Organizations (Organization +)	
Media (M)	Report	
Public Health (PH)	Clinical, Disease, Emergency/Emergencies (Emergency +), Outbreak, Patient/Patients/Case (Patient +), Vaccine/Vaccines/Vaccination (Vaccine +), Protect/Protective/Protection/Protecting (Protect +), Provide/Supply/Available (Provide +), Testing	
Symptoms/Symptom (S)	Fatigue	
Territory (T)	China, European, Global, USA, World, Wuhan	

Figure 2 YoY +% and BI values of subordinate hot words.

Temporal distribution of subordinate hot words and domains

In accordance with statistics from WHO and BI, we concluded the occurrence frequencies and corresponding peaks (2019/12/01–2021/11/30) of subordinate hot words in Table S4. In curves of cases and deaths, we found the two peak periods (2020/01–2020/04 and 2021/05–2021/07) (Reis & Brownstein, 2010) and secular distribution tendency of irregular fluctuation (Fig. 3). The subordinate hot words frequencies were also mainly concentrated in the two peak periods (P value: 0.0001), proving that hot words efficiently tracked the scale of SARS-CoV-2 infections through online channels and consistently gave feedback in a timely manner. Hence, cases and deaths both functioned as variables for analyzing temporal correlation, though both had a significantly varying orders of magnitude. Besides, a few subordinate hot words dispersedly occurred in other periods, while their impact on correlation could be ignored.

Figure 3 Temporal distribution of subordinate hot words, confirmed cases and deaths (P value: 0.0001).

BI values of domains were ranked in order as follows: CR (522614) >T (102594) >PH (4409) >S (1959) >G (991) >M (271). Therefore, subordinate hot words in CR and T domains had the ability to capture the instant message of COVID-19 (P value of BI values difference between domains: 0.6619) (Fig. 4) in comparison with the other domains. Notably, CR and T domains also focused on the periods of COVID-19 outbreak periods (2020/01–2020/04 and 2021/04–2021/07) and the China’s Spring Festival holiday in 2021(2021/01–2021/03). Rising mobility during holiday periods would promote the spread of SARS-CoV-2 (Chen et al., 2020), and CR and T domains forecasted COVID-19 severity, observed the persistence and periodicity of outbreaks and surveilled potential outbreaks. G domain was dispersedly distributed in eleven months of 2020 and 2021 without apparent temporal correlations. Hence, this domain lacked sensitivity of real-time or large-scale surveillance for COVID-19 and was vulnerable to policies. For the other domains (PH, S and M), they were also mainly distributed in outbreak periods (2020/01–2020/04 and 2021/04–2021/07).

Figure 4 Temporal distribution of domains in 2020 and 2021 (P value of BI values difference between domains: 0.6619).

Gender distribution of domains and subordinate hot words

According to the gender inquiry service provided by BI, we summarized the gender distribution of domains and calculated the standard deviation (SD) of subordinate hot words within their corresponding domains. The detailed gender distribution of subordinate hot words was presented in Figs. S1 and S2. Overall, males and females held 51.90% and 48.10% of search users respectively, and there was no obvious gender bias in users (Fig. 5). T domain (67.69%) and S and PH domains (57.95%, 56.61%) were searched more by males and females respectively. Therefore, males relied on public services related to geographic information (e.g., international news and epidemic data), while females paid more attention to medical and healthcare information (e.g., medical supplies). CR (SD: 11.41%), M (SD: 0) and G (SD: 5.64%) domains had implicit gender bias due to the differences between the male and female proportions with 10.6%, 7.08% and 1.54% respectively. Except CR domain, M and G included classic attributes of social life, meaning that COVID-19 have not affected the fulfillment of social needs and the normal order of social life between different gender groups (Lai et al., 2020).

Figure 5 Gender distribution of domains and subordinate hot words (SD: Standard deviation (%)).

Age distribution of domains and subordinate hot words

Users were divided into five age groups: ≤19 years, 20–29 years, 30–39 years, 40–49 years and ≥50 years. We summarized the SD of age distribution about each subordinate hot word (Fig. S3 and Table S4) and the average SD within each domain (Fig. 6). In Fig. 6, the average proportions of age groups were 10.48% (≤19), 28.01% (20-29), 37.93% (30–39), 15.73% (40–49) and 7.85% (≥50) respectively. Individuals aged 20 to 39 years (65.94%) were the main search service users, and those aged under 19 years and over 50 years had a lower online search intensity than other groups. Individuals aged 20 to 29 years and 30 to 39 years focused more on M domain (45.43%, SD: 15.37%) and S domain (51.66%, SD: 16.59%) respectively, thus above age groups possibly changed their attitudes towards COVID-19 especially when they received various media forms of COVID-19 news or symptoms-related information. Individuals aged 40 to 49 years had no preference for any domains. The proportions of different age groups who searched S and T domains were the highest (SD: 16.59%) and the lowest (SD: 8.87%) respectively. Therefore, geographical information related to COVID-19 was more likely to be concerned and accepted by various age groups.

Figure 6 Age distribution of domains (SD: Standard deviation (%)).

Geographical distribution of subordinate hot words based on parameter estimation

The number of confirmed cases in provinces lay within the range of 1 (Tibet province) to 68,311 (Hubei province) as of November 30, 2021 (Table 3), and that in 14 provinces surpassed 1000. We found that confirmed cases in all provinces were best fitted by the Exp3P2 (Exponential function whose exponent is a second order polynomial with three parameters) model (adjusted R2: 0.99119, P value <0.05) (Fig. 7). The normalized GDCV of provinces ranged from 26.7765 (Tibet province) to 31.0000 (Hubei province) (Table 3). The magnitude difference in the GDCV of provinces was significantly smaller than that in case numbers, thus, GDCV was conducive to obtaining confidence intervals. The confidence interval was computed (Table 4) according to the sum of GDCV of subordinate hot words (P value <0.05) (Table S5). According to the above confidence interval, the potential frequencies (122512 combinations) of COVID-19 outbreaks in provinces were finally estimated in Table 3 and Table S6.

Table 3 Confirmed cases, GDCV and occurrence frequencies of provinces (Deadline: 2021/11/30) (Except Hongkong, Macao and Taiwan).

Serial numbers	Provinces	Cases	GDCV	Frequencies	
31	Hubei	68,311	31.0000	135,668	
30	Guangdong	3,279	29.9642	102,310	
29	Shanghai	2,824	29.9115	94,630	
28	Heilongjiang	1,992	29.7874	77,620	
27	Jiangsu	1,619	29.7240	69,238	
26	Yunnan	1,668	29.7170	68,340	
25	Henan	1,636	29.7133	67,837	
24	Zhejiang	1,501	29.6861	64,373	
23	Hebei	1,453	29.6744	62,967	
22	Fujian	1,319	29.6396	58,582	
21	Sichuan	1,266	29.6248	56,740	
20	Hunan	1,197	29.6046	54,353	
19	Beijing	1,191	29.6028	54,139	
18	Shandong	1,011	29.5435	47,123	
17	Anhui	1,008	29.5424	47,006	
16	Xinjiang	981	29.5326	45,875	
15	Jiangxi	959	29.5243	44,963	
14	Liaoning	775	29.4468	36,445	
13	Shaanxi	705	29.4122	32,835	
12	Inner Mongolia	618	29.3639	27,941	
11	Chongqing	610	29.3591	27,471	
10	Jilin	582	29.3418	25,821	
9	Tianjin	528	29.3060	22,407	
8	Guangxi	381	29.1852	12,623	
7	Gansu	344	29.1472	10,249	
6	Shanxi	264	29.0483	5,433	
5	Hainan	190	28.9243	2,101	
4	Guizhou	159	28.8568	1,167	
3	Ningxia	122	28.7557	409	
2	Qinghai	30	28.3684	0	
1	Tibet	1	26.7765	0	

Figure 7 Nonlinear curve fitting of the number of cases.

Table 4 The parameters of confidence interval based on the GDCV of subordinate hot words.

n a	1-α b	S c	x¯ d	t α/2 e	
31	0.95	0.5932	298.8680	2.0639	
Notes.

a n: Sample size of hot words.

b 1- α: Confidence level.

c S: Corrected standard deviation of samples.

d x¯: The average sum of GDCV of subordinate hot words.

e tα/2: Critical value of T distribution.

The provinces and administrative divisions in mainland China were divided into five early-warning levels: potential frequencies >100,000, level I early-warning regions (EWRs I), potential frequencies >50,000, EWRs II, potential frequencies >10,000, EWRs III, potential frequencies >1,000, EWRs IV, potential frequencies <1,000, EWRs V (Fig. 8). EWRs I and II centered around the Central region (Hubei, Hunan and Henan provinces), and the early-warning levels fell around gradually towards the surrounding provinces and geographical regions (EWRs III and IV). Some provinces in the coastal regions (the East and South regions) with a dense population and heavy traffic also belonged to EMRs I and II, and would even suffer repeated COVID-19 outbreaks in the future due to international transport. Because the Northwest region (Ningxia, Qinghai and Tibet provinces, etc.) located in plateau or desert areas had a very low population density and traffic (Cai et al., 2021), provinces in this region would be all EMRs III and V. Besides, the EMRs II in the Northeast and Southwest regions were all inland border provinces (Russia and Southeast Asia) (Geology, 2022) and were first affected by the international epidemic emergencies.

Figure 8 Levels of early-warning provinces and Chinese geographical divisions.

Discussion

Principal findings

Our findings demonstrated that hot words promptly and efficiently monitored and reflected the impact of COVID-19 on the people in different age groups, gender groups and regions and their behavior patterns. Therefore, government health management departments could adopt the necessary measures based on the attitudes of different groups to COVID-19. Researches on users’ information-seeking behavior have confirmed that the big data and national databases adeptly organized online browsers’ behavior information and integrate their behavior profiles for monitoring potential epidemic outbreaks and activities of social groups (Dreher et al., 2018; Xiang et al., 2020). Compared with statistics from medical institutions and CDC, online information was more convenient, comprehensive and detailed, and the public favored a preliminary consult online before seeking medical help (Davtyan, Brown & Folayan, 2014; Nimavat et al., 2021). Besides, hot words depended on real-time online data were immune from “official blackout periods”, and users took appropriate measures against possible COVID-19 outbreaks by using reliable information such as hot words.

Based on a temporal fitting with 3 dependent variables (Fig. 4), we confirmed that subordinate hot words and domains could effectively analyze the current epidemic situation and indirectly reflected attitudes taken by citizens towards the epidemic (Chen et al., 2020). Springs and winters (defined by National Meteorological Administration of China) (Wei et al., 2021) were high-occurrence seasons for epidemics, while COVID-19 appeared not only in the spring of 2020 (2020/03-2020/05) but also in May 2021. Firstly, affected by real-time policies for the prevention and control taken by the Chinese government (Chen et al., 2020), COVID-19 had been under timely control after the Spring Festival in 2020, and thus only a few imported cases from abroad might lead to small-scale outbreaks even in the winter of 2020 (Shen et al., 2020). Secondly, traffic flows on holidays (May Day) might also lead to the outbreak of COVID-19 (Chen et al., 2020). Therefore, hot words performed quickly and accurately in monitoring the epidemic situation and short-term forecast, while for secular forecast of hot words it is susceptible to China’s actual condition (policies) and holidays instead of seasonal factors. Additionally, CR and T domains were instrumental in stably monitoring the epidemic situation, and similar views have emerged in previous studies (Li & Liu, 2020; Rader et al., 2021; Wynants et al., 2020). As the most non-temporal correlation domain, G domain could only prove the Chinese government’s positive attitude towards the epidemic (Aravindhan et al., 2021), and was not used as a characteristic domain for forecasting.

Overall, the subordinate hot words of T, PH and S domains effectively reflected whether the epidemic had a serious impact on male and female groups, thereby developing prevention and control measures based on physiological differences (physical agility, childbirth and spiritual anxiety) between men and women (Alahdal, Basingab & Alotaibi, 2020; Wang et al., 2020a; Wang et al., 2020b). The proportion of females who searched for the words in S (Symptoms/Symptom, 57.95%) and PH (Public health, 56.61%) domains related to COVID-19 was much greater than that of males. According to the data from Lai et al. (2020), above 80% cases was the female population in Wuhan and Hubei province outside Wuhan, and the proportion of outside Hubei province was 60%, which indicated that the gender distribution of domains qualitatively forecasted the potential proportion of male to female cases. Need to add that, though male and female population were most concerned about “Global” in T domain and “Outbreak” in PH domain respectively (Figs. S2 and S3), while the accuracy of testing results of a single subordinate hot word was far less stable than that of a domain. Therefore, domains are more convincing than a single subordinate hot word as a basis for forecasting gender distribution.

The population aged below 20 years lacked fundamental understanding of the epidemic situation, and the epidemic situation and lockdown caused older population greater feelings of loneliness (Alahdal, Basingab & Alotaibi, 2020), so these groups adopted a passively accepted attitude towards external information. A further cause of the group aged over 50 years accounting for a low proportion of search users was that the proportion of internet users in this age group was only 26.8% by the end of 2020 (CNNIC, 2021). Medical institutions necessarily furnished the above people with some extra assistance. The middle-aged population (20–49) was actively concerned about epidemic information due to social contact and families (Alahdal, Basingab & Alotaibi, 2020). The latest data revealed that individuals aged 20 to 29 years depended on the media to realize current epidemic trend, while individuals aged 30 to 49 are more likely confirmed patients (Lai et al., 2020). Hence, the analysis objects of age distribution mainly concentrated in the 20–50 age group. We finally selected G, PH and T domains with higher SD values (SD: 21.86%, 26.95% and 17.27%) to characterize the group of 20–29, 30–39 and 40–49 (age distribution: 30.21%, 40.06% and 18.02%) years old for improving the accuracy of forecast. In other words, the search volume of subordinate hot words in G, PH and T domains might rise notably before the above three age groups would be infected on a large scale, and which assisted government agencies to carry out medical assistance for people of different ages in a planned way. Moreover, S and M domains were not given priority to epidemic surveillance, because only one subordinate hot word (SD: 0) in these domains.

The results of geographical distribution showed that hot words were beneficial to the monitoring of the epidemic situation to a certain extent. Regional search trends of COVID-19 were consistent with those of other epidemiological studies (Wang et al., 2020a; Wang et al., 2020b; Wei et al., 2021; Xiang et al., 2020). Central China region always ranked first in hot word search, followed by North, South and East China regions, and Southwest, Northeast, Northwest China regions were lowest in it. Wuhan (Central China region) as the earliest and most serious epidemic outbreak city had been effectively blocked through government agencies, resulting in outlier data about the cases of Central China region (Lai et al., 2020; Mo et al., 2020). Due to the same regional distribution tendency between total cases and search volume, we chose student’s t-test model (P < 0.05) which had excellent predictive power to quantitatively estimate provinces where the epidemic would break out in the future (De Muth, 2009). Those living in areas with fewer economic and technologic advantages (CNNIC, 2022; STATS, 2021) used online services less, causing the lack of their information records on BI. Nevertheless, the rankings of geographical distribution of hot words search volume only presented the top 10 provinces, so the early-warning levels we evaluated (Fig. 8) was hardly impacted by BI (especially for Ningxia, Qinghai and Tibet provinces et al. with lower search frequencies of hot words, few confirmed cases and lower internet penetration rate (Table 3)) (CNNIC, 2022; Zhong et al., 2022). Similar researches on geographical distribution of BI influenced by the internet penetration rate were inclined to adopt provinces and Chinese administrative divisions as the basic geographic units for declining the effect of counties or towns with low internet usage on geographical distribution (Fang et al., 2021; Wei et al., 2021).

Comparison with prior works

The crucial criticisms of the infodemiology findings conducted by the data analysis platform based on the typical netizen behavior data (e.g., Google trends and Baidu Index) revealed that the platform data provided inaccurate information and designed algorithms with logical errors, resulting in a large error compared with the actual situation (Mavragani & Ochoa, 2019; Wei et al., 2021). Therefore, most researchers employed the characteristic words of certain infectious diseases and the methods of removing noise words to enhance the accuracy of the results (Huang et al., 2020; Dreher et al., 2018; Mavragani & Ochoa, 2019). This paper introduced the concept of “hot words” and selected the high-frequency words collected from the real-time infectious disease information platforms as the characteristic words to search in BI. At the same time, we designed two-rounds of hot words screening and double-level hot words classification, which not only strengthened the accuracy of the results, but also excluded the vast majority of noise words and COVID-19-unrelated words. Compared with the monitoring models focusing on time series analysis (Xiang et al., 2020), our research also provided gender, age and geographical information of the public for constructing demographic portraits from different dimensions, so as to take disease control measures for different groups. Additionally, this paper did not implement complex statistical algorithms such as dynamic regression models (Mishra et al., 2019), but used the basic algorithms with strong repeatability, and took the domains and subordinate hot words as dynamic parameters to adapt to the epidemic monitoring under the characteristics of different times and populations.

Strengths

Inspired by WHO keywords classification of epidemic news (WHO, 2021) and the binary classification methods (symptoms terms—symptoms equivalent terms) proposed by Wei et al. (2021), our dissertation extracted infodemiology features which were named as “Domains” to constitute a double-level classification system from all subordinate hot words. These domains with social (M, G and PH domains) or epidemic (S, CR and T domains) characteristics were convenient for personnel in different research fields to study the epidemic situation. Meanwhile, the introduction of domains could be well compatible with some subordinate hot words that lack BI value data, so as to merge the BI values of other subordinate hot words into them. For some hot words such as “Coach” which was one of the symptoms of COVID-19 (Pei, De Vries & Zhang, 2021), their BI values recorded in COVID-19 outbreak periods were even higher than that in free periods, so this study first took YoY +% (Pei, De Vries & Zhang, 2021) to exclude them from the final hot words table. After multiple attempts based on the results of the initial round of screening hot words (Table S3) to set the threshold sizes of YoY +%, we took 5% (Heald et al., 2021) as the threshold of YoY +% to screen epidemic-unrelated hot words and preserve enough epidemic-related hot words.

Because BI values were weighted values which hardly reflected the search volume of users for current affairs and policies (Wang et al., 2020a; Wang et al., 2020b), our studies replaced BI values with occurring frequencies (peaks) of hot words to establish the temporal correlation of hot words and the epidemic severity. Referring to the methods of defining peaks within a long-term time series (2019/12/01–2021/11/30) applied by Xiang et al. (2020), “kurtosis >3” was applied as the peaks of hot words to maintain the accuracy of all word frequencies. Besides, BI mined users’ search data of the whole network according to the basic information of registered Baidu platform users, and further clustered the population by the searched keywords, so as to find the gender and age distribution of the user groups queried by hot words (Qiu et al., 2020). For example, if a user searched for women’s products much more than men’s products, this user was labeled as a woman, or if a user searches for health care products for the elderly much more than for school or work supplies, this user was divided into the group over the age of 50. Though BI provided hot words search rankings at the regional and provincial levels collected from the non-current resident IP address of users, while the BI values of provinces or regions was not available in BI. Therefore, we compiled the confirmed cases of each province as quantitative data and hot words as qualitative data for recording the search rankings of each province to carry out geographical distribution research.

Official COVID-19 statistics were successively gathered by sanitation monitor management systems and epidemic prevention medical stations at local and national levels (Pan et al., 2021; Wei et al., 2021). This policy was indeed conducive to monitoring of national epidemic situation based on big data, but the process of data collection consumed a large amount of manpower and material, but also to cause non-timely remediation and control of some potential outbreaks during the process of summarizing data. The methods mentioned in our study could simply forecast COVID-19 outbreaks and the public behavior profiles (Jung et al., 2020). Except forecasting geographical distribution by the parameter estimation method, temporal, gender and age distribution were analyzed by appropriate correlation tests. We ultimately discovered that CR and T domains could monitor severity of epidemic situation in real time; T, PH and S domains reflected whether the epidemic had a serious impact on male and female groups; G, PH and T domains could characterize the group of 20–29, 30–39 and 40–49 years old when they were menaced by COVID-19. When COVID-19 would be soon to break out, above BI values of domains and subordinate hot words would increase dramatically or begin to appear more obvious fluctuation. Therefore, the information retrieved from BI was published faster than that from Chinese government agencies, and domains and subordinate hot words could monitor and even forecast the potential or spreading epidemic situation. More importantly, besides the government health care institutions, the mass media and the public expediently analyzed the current and even prospective epidemic situation by this method.

Limitations

In this article, although the monitoring methods based on authoritative big data platforms could effectively respond to potential outbreaks, several unsolved limitations were still discussed. Effected by the emergence of Omicron mutant strains (Vaughan, 2021) and Chinese government’s epidemic control policies (Wang et al., 2020a; Wang et al., 2020b), models established by the existing data were inevitably affected, which meant that these models only attained desired results in real-time monitoring and short-term forecast. CCDC and BI could not provide private information (educational background, nationalities and socioeconomic status) and time characteristic curves of gender, age and geographical distribution. Furthermore, some COVID-19 related hot words, such as “Muscle or body aches” and “Doses”, were not input into public databases and lack corresponding epidemiological data.

All monitoring methods or models had timeliness and were vulnerable to outliers (the number of early confirmed cases in Wuhan) (Hossain et al., 2020). However, Wuhan, as the most developed city with migrating people, a large population base and a high cargo throughput in Central China region, was likely to be impacted again by the epidemic (Li et al., 2021). Therefore, we integrated all information on populations into models instead of being restricted to specific factors. We also designed reliable algorithms to screen hot words and analyze the temporal, gender, age and geographical distribution of different groups for reducing decision-making risks. What’s more, in terms of information associated to the epidemic situation, Baidu platform should strive for government permission and support, revise and add keywords, and timely supplement real-time data. Thus, an in-depth demographic investigation could be carried out in future, and regional medical and health departments could get services in time.

Conclusions

The objective of this article was to investigate the feasibility of summarizing the users’ behavior profiles based on hot words and BI. We found that domains and subordinate hot words were effective in instantly monitoring the impact of COVID-19 on different groups and forecasting COVID-19 epidemic in short time through the verification of time series, gender, age and geographical distribution. Compared with other research on the epidemic situation, algorithms used to screen hot words in our study were more intelligible, convenient and flexible, and our results were highly consistent with the facts. Therefore, the government and medical professionals could refer to our analytical processes and results for formulating applicable policies and making real-time measures to monitor the epidemic situation. In the future, we will continue to improve the algorithms of the models and incorporate more parameters that are probably associated with the epidemic situation for reference.

Supplemental Information

Figure S1 Gender distribution of subordinate hot words (Male population)

Click here for additional data file.

Figure S2 Gender distribution of subordinate hot words (Female population)

Click here for additional data file.

Figure S3 Age distribution of subordinate hot words

Click here for additional data file.

Table S1 Home and subpages of databases

Click here for additional data file.

Table S2 Word frequencies based on Python codes

Click here for additional data file.

Table S3 The bilingual table (English and Chinese) and YoY+% of hot words

Click here for additional data file.

Table S4 Temporal and age distribution of hot words

Click here for additional data file.

Table S5 The GDCV of subordinate hot words

Click here for additional data file.

Table S6 All possible occurrence frequencies of COVID-19 outbreaks in provinces

Click here for additional data file.

Supplemental Information 10 The python codes to sort hot words by word frequency

Click here for additional data file.

Supplemental Information 11 The python codes to get all provinces with potential outbreaks based on the confidence interval

Click here for additional data file.

Abbreviations

BI Baidu Index

BIDA Baidu Index daily average

COVID-19 Corona virus disease 2019

CR domain COVID-19 Related domain

EWR Early-warning region

G domain Government domain

GDCV Geographical distribution characteristic values

M domain Media domain

PH domain Public Health domain

SARS-CoV-2 Severe acute respiratory syndrome coronavirus 2

S domain Symptoms/Symptom domain

SD Standard deviation

T domain Territory domain

YoY +% Year-on-year growth rate

Additional Information and Declarations

Competing Interests

Author Contributions

Data Availability

The authors declare there are no competing interests.

Shuai Jiang conceived and designed the experiments, performed the experiments, analyzed the data, prepared figures and/or tables, authored or reviewed drafts of the article, and approved the final draft.

Changqiao You conceived and designed the experiments, performed the experiments, analyzed the data, prepared figures and/or tables, authored or reviewed drafts of the article, and approved the final draft.

Sheng Zhang analyzed the data, prepared figures and/or tables, authored or reviewed drafts of the article, and approved the final draft.

Fenglin Chen analyzed the data, prepared figures and/or tables, authored or reviewed drafts of the article, and approved the final draft.

Guo Peng analyzed the data, prepared figures and/or tables, authored or reviewed drafts of the article, and approved the final draft.

Jiajie Liu analyzed the data, prepared figures and/or tables, authored or reviewed drafts of the article, and approved the final draft.

Daolong Xie analyzed the data, prepared figures and/or tables, authored or reviewed drafts of the article, and approved the final draft.

Yongliang Li conceived and designed the experiments, prepared figures and/or tables, authored or reviewed drafts of the article, and approved the final draft.

Xinhong Guo conceived and designed the experiments, prepared figures and/or tables, authored or reviewed drafts of the article, and approved the final draft.

The following information was supplied regarding data availability:

The code and data are available in the Supplemental Files.

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
