# Peer review of "Using search trends to analyze web-based users’ behavior profiles connected with COVID-19 in mainland China: infodemiology study based on hot words and Baidu Index"

_PeerJ, doi:10.7717/peerj.14343_

## Round 0.1 · original submission · Major Revisions

This manuscript provided useful information and well-written. However, it can be further improved based on the comments from the reviewers.

Reviewer 1 ·

Basic reporting

This is an interesting and well-written manuscript. The authors summarized web-based users’ behavior profiles based on “hot words” and performed analysis for temporal distribution and demographic portraits of COVID-19. The study is well-designed and the method adopted to perform the analysis are appropriate and reasonable. The study results are clearly explained and interpreted. A few specific comments and questions.

1. In the abstract, the authors used a few abbreviations, like CR, PH, etc. Without reading the main text, the readers have no clue about what they mean. Suggest to spell them out in the abstract to avoid confusing readers.

2. In Line 243, the authors mentioned that they used an exponential model to fit the non-linear curve of confirmed cases. However, In Figure 6, is Hubei an outlier here?

3. In the Discussion section, the authors mentioned the domains and subordinate hot words were effective in forecasting COVID-19 epidemic in short time. In my view, the analysis performed this study is more of in a descriptive and confirmative nature. The correlation analysis the authors did is valuable, but it’s not clear in the article how people can use the analysis in this article to forecast epidemic in the future. It would be helpful for authors to comment and elaborate on it.

Experimental design

See comments above

Validity of the findings

See comments above

Reviewer 2 ·

Basic reporting

Thank you for providing me with a chance to review this article. The authors investigated the attitudes of internet users in China towards the Covid-19 pandemic by mining hot words and collecting Baidu Index data. In addition, this article also looked into the temporal, age, and gender distribution of hot word search, as well as forecast future spatial distribution of the Covid-19 pandemic in China. Finally, the authors concluded that domains and hot words effectively monitored the impacts of Covid-19 on diversified groups and forecasted the Covid-19 pandemic. Overall, this paper is interesting but can be improved.

1. I recommend avoiding the use of inessential abbreviations so that the readers don't have to repeatedly locate the table of abbreviations or go back to the earlier sections.
2. Also, it is not necessary to use acronyms in the abstract.
3. Line 135 "... was COVID-19-unrelated when ...": do you mean "related"?
4. Line 144: "... the above 'question'": do you mean "problem"/"issue"?
5. Line 145-148: This sentence is long, and I found it uneasy to follow.

Experimental design

6. The purpose of the study needs to be crisply stated in the introduction section.
7. Line 161-170: I found it hard to understand the rationale behind the model (why it can forecast future outbreaks). More details of the model or a flowchart might help readers understand how the model works.

Validity of the findings

8. Line 67-68: the Internet penetration rate in China was not as high as nearly 100%, according to World Bank (https://data.worldbank.org/indicator/IT.NET.USER.ZS?locations=CN). People in central, western, and rural China may have lower access to the Internet.
9. Line 37-38 / Line 230-231: it's likely that middle-aged people had higher search intensity because more middle-aged people are using the Internet in China, not because they paid more attention to the epidemic.

Reviewer 3 ·

Basic reporting

The article is well-written in English, however, there are several grammatical errors to be corrected.

Information included in the literature is sufficient and relevant.

Line 620: The reference used might not reflect the actual internet penetration rate in non-urban area. Kindly validate this.

Experimental design

The research questions are well defined and relevant. However, the overall objective of the study is not stated clearly in the abstract and main text.

In this study, authors have built a COVID-19 monitoring model based on the big-data platforms. The output of this research is significant especially to the policy maker. It will be more meaningful if an in-depth demographic investigation could be carried out in future to include data such as socioeconomic status, ethnicity or educational background.

Validity of the findings

It is undeniably true that infodemiology platform provides a real-time and faster way to investigate the impact of COVID-19 to public, before a national surveillance and epidemiological reporting system are well established. However, several points needed to be clarify and further evaluated.

Firstly, although Baidu is the most widely used platform and monopolized the searching request in mainland China, the users; shift to social media for searching is rising. Thus, data from these growing social media sites might need to be considered to prevent bias.

Next, according to the statistic provided by the China International Network Information Center (https://www.cnnic.net.cn/gywm/xwzx/rdxw/20172017_7086/202208/t20220831_71823.htm), by June 2022, the Internet penetration rate was 74.4%, which is different from the 100% as mentioned in the manuscript. This suggested that BSI might not be able to cover certain areas. Therefore, the accuracy of this model needs further evaluation.

Annotated reviews are not available for download in order to protect the identity of reviewers who chose to remain anonymous.

---

## Round 0.2 · accepted · Accept

I have reviewed the revised manuscript and happy to see that all the comments are addressed.

Reviewer 2 ·

Basic reporting

I thank the authors for their work on the revision. The revised manuscript properly addresses all the points raised in my earlier review.

Experimental design

The revised manuscript properly addresses all the points raised in my earlier review.

Validity of the findings

The revised manuscript properly addresses all the points raised in my earlier review.

Reviewer 3 ·

Basic reporting

Most of the grammatical errors have been corrected.

Experimental design

Overall objective of the study was well defined.

Validity of the findings

No comment